# Quantitative Analysis of the Potency of Equimolar Two-Drug Combinations and Combi-Molecules Involving Kinase Inhibitors In Vitro: The Concept of Balanced Targeting

**DOI:** 10.3390/ijms22179569

**Published:** 2021-09-03

**Authors:** Suman Rao, Benoît Thibault, Lisa Peyrard, Anne-Laure Larroque-Lombard, Martin Rupp, Cédric Thauvin, Bertrand J. Jean-Claude

**Affiliations:** Cancer Drug Research Laboratory, Department of Medicine, Division of Medical Oncology, The Research Institute of the McGill University Health Center/Glen Hospital, Montreal, QC H4A 3J1, Canada; sumanrao5664@gmail.com (S.R.); peyrardlisa@gmail.com (L.P.); anne-laure.larroque@muhc.mcgill.ca (A.-L.L.-L.); ruppmartin89@gmail.com (M.R.); cedricthauvin@gmail.com (C.T.)

**Keywords:** equimolar combinations, hybrid molecules, combi-molecules, balanced targeting, kinase inhibitors, multi-targeting

## Abstract

The median-effect principle proposed by Chou and Talalay is the most effective approach to parameterize interactions between several agents in combination. However, this method cannot be used to evaluate the effectiveness of equimolar drug combinations, which are comparative references for dual-targeting molecular design. Here, using data acquired through the development of “combi-molecules” blocking two kinases (e.g., EGFR-c-Src and EGFR-c-Met), we established potency indices for equimolar and dual-targeted inhibitors. If the fold difference (κ) between the IC50 of the two individual kinase inhibitors was >6, the IC50 of their equimolar combination resembled that of the more potent inhibitor. Hence, the “combi-targeting” of the two kinases was considered “imbalanced” and the combination ineffective. However, if κ ≤ 6, the IC50 of the combination fell below that of each individual drug and the combi-targeting was considered “balanced” and the combination effective. We also showed that combi-molecules should be compared with equimolar combinations only under balanced conditions and propose a new parameter Ω for validating their effectiveness. A multi-targeted drug is effective if Ω < 1, where Ω is defined as the IC50 of the drug divided by that of the corresponding equimolar combination. Our study provides a methodology to determine the in vitro potency of equimolar two-drug combinations as well as combi-/hybrid molecules inhibiting two different kinase targets.

## 1. Introduction

The implication of several signaling proteins in a complex network of signal transduction pathways is a commonly occurring event in advanced cancers. These signaling interactions between growth factor receptors (the epidermal growth factor receptor (EGFR), hepatocyte growth factor receptor (c-Met), platelet derived growth factor receptor (PDGFR), vascular endothelial growth factor receptor (VEGFR), etc.) and cytoplasmic non-receptor tyrosine kinases and transcription factors (c-Src, c-Abl, JAKs, STAT3, β-catenin, etc.) not only synergize to promote tumor growth, survival, and metastasis, but also mediate resistance to targeted therapies through the activation of compensatory signaling pathways [1,2,3]. Thus, in recent years, strategies designed to overcome resistance mediated by compensatory signaling have involved the use of a multi-targeted approach [4,5]. Within this context, over the past decade, we developed a novel approach termed “combi-targeting” that sought to design agents designated as “combi-molecules” capable of inducing tandem blockade of two divergent biological targets (e.g., EGFR, PARP, MEK, and DNA) [6,7,8,9,10,11,12,13,14,15,16]. Further work on the concept led to the synthesis of molecules rationally designed to target two oncogenic tyrosine kinases involved in adverse signaling [17,18]. More specifically, we demonstrated the feasibility of combi-molecules capable of blocking c-Src and EGFR as intact molecules and further degrading to two intact inhibitors of the two targets. Such types of molecules capable of behaving as dual-targeting agents, while being a prodrug of two active inhibitors, were designated as type III targeting molecules. This designation was chosen to distinguish them from their type I and type II predecessors [19]. Type I combi-molecules are designed to block only one target as an intact molecule and require hydrolysis to be able to block their secondary target [8,20,21,22]. Type II combi-molecules are dual-targeting molecules that do not require hydrolysis to hit their two targets [23,24]. The targeting mode that is referred to as type II is the most commonly used approach in the literature and the resulting molecules are often referred to as hybrid or chimeric molecules [25,26]. Regardless of how they are referred to, combi- or hybrid molecules are carriers of two or more equimolar agents generating two or more distinct effects, each of which being associated with one of the moieties of the parent molecule. Thus, their potency is often evaluated in comparison with equimolar combinations of agents acting by the same mechanisms of action [27,28,29].

Recent efforts toward inducing a tandem blockade of multiple signaling pathways have increased interest in the use of equimolar combinations. We and others have frequently used them as a reference for studying the biological effects of newly designed hybrid molecules [18,22,29,30,31]. Despite the extensive use of equimolar combinations as a reference for multi-targeted drugs and the current interest in combinations of targeted agents, little is known about criteria to define the magnitude of potency of mixtures of two drugs administered in vitro in an equimolar combination modality.

In the past, Chou and Talalay [32] demonstrated that the type of interactions between two or more drugs combined in an equieffective ratio could be parameterized by the median-effect principle, whereby the combination index (CI) is used to define synergy when the value is < 1, additivity when the value is 1, and antagonism when the value is > 1 [32,33,34]. However, such calculations cannot be performed under conditions where the drugs are combined in an equimolar ratio, a condition that is best suited for comparisons with hybrid drugs or combi-molecules [29,30,35,36]. In this study, using current and past data acquired from our kinase–kinase and kinase–DNA targeting programs, we propose a quantitative model based on simple mathematical equations for determining the degree of effectiveness of an equimolar drug combination. We also propose criteria where equimolar drug combinations or hybrid/combi-molecules would be an effective targeting strategy.

The approach we chose to study was to target tyrosine kinases such as c-Src, c-Met, and EGFR, which are known oncogenes driving various tumors through complex signaling interplay. Here, we first determined the IC50 values for growth inhibition induced by their clinical inhibitors both as single agents and equimolar combinations against human cancer cell lines of various histological origins. Subsequently, we analyzed the trends of the IC50 values of these combinations in comparison with individual drugs as well as combi-molecules synthesized in our lab, targeting EGFR, c-Src, and c-Met [19]. “Imbalanced targeting” was noted if the fold difference (κ) between the IC50 of the two individual drugs was above 6, resulting in the IC50 of their equimolar combination to be similar to that of the most potent of the two drugs. On the contrary, “balanced targeting” was observed if the fold difference (κ) between the IC50 of the two drugs was less than 6, resulting in the IC50 of their equimolar combination being less than the IC50 of either drug. Likewise, we propose that in vitro, a hybrid or combi-molecule can be considered effective in a given cell system when its IC50 for growth inhibition is lower than or equal to that of a balanced targeting combination.

## 2. Results

### 2.1. Growth Inhibitory Potency of Single Versus Equimolar Combinations of Clinical Inhibitors on a Panel of Cancer Cell Lines

#### 2.1.1. EGFR-c-Src Targeting

In order to profile the responses, we primarily screened a panel of cancer cell lines using single and equimolar combinations of clinically approved tyrosine kinase inhibitors (TKIs) targeting kinases engaged in synergistic crosstalk, and further extended the screening to kinase–DNA targeting drug combinations. The cell lines used in this study included breast, lung, prostate, ovarian, head and neck, and brain cancer cells along with the NIH3T3 panel of wild type, EGFR, and Her2 transfected cell lines. Clinical TKIs including gefitinib (EGFR inhibitor), crizotinib (c-Met inhibitor), dasatinib (c-Src inhibitor), and the DNA alkylating agent temozolomide were used (Figure 1a). In all the cell lines, the range of IC50 for dasatinib varied from 0.01 ± 0.0003 μM to 16.4 ± 5.2 μM, whereas those for gefitinib varied from 0.22 ± 0.01 μM to as high as 73.7 ± 7.7 μM, except in the EGFR TKI-sensitive HCC827 cell line with a deletion in exon 19 in the EGFR kinase domain (Del E746-A750), where it showed an IC50 value in the nanomolar range (0.003 ± 0.0005 μM). Importantly, when the two drugs were combined (gefitinib + dasatinib) in an equimolar manner in this cell line, the IC50 of the combination fell in the range of that of dasatinib. It is noteworthy that in HCC827 cells that showed an IC50 for dasatinib 30-fold less than that of gefitinib, the IC50 of the combination was in the same range as that of the latter. The IC50 values of the gefitinib + dasatinib equimolar combination fell below that of each individual drug only in MDA-MB-468, 22RV1, UM22A, A2780, and NIH 3T3 cells wherein dasatinib and gefitinib exhibited IC50 values in the 1–4-fold difference range (Figure 1a). The overall response profiles are depicted in the average graphs as shown in Figure 1b–g and calculated as the IC50 of the drug in each cell line minus the average IC50 of the drug in the entire panel of cell lines. Response profiles were calculated for gefitinib, dasatinib, and the equimolar combination of the two for cell lines, exhibiting IC50 values of gefitinib and dasatinib in different or similar ranges. As can be seen, the response profile of gefitinib + dasatinib (Figure 1d) more closely resembled that of dasatinib (the drug that is 6-fold more potent than gefitinib) (Figure 1b–d). By contrast, in the cell panel in which gefitinib and dasatinib showed response profiles of similar magnitude, the combination of gefitinib + dasatinib (Figure 1g) appeared to yield IC50 values in a lower range than that of each individual profile, indicating enhanced potency when compared with each drug alone (Figure 1e–g). This represents what we define as “balanced targeting”, which is a sharp contrast with profiles b–d, which clearly exemplify a case of “imbalanced targeting”.

#### 2.1.2. EGFR-c-Met Targeting

Further studies targeting c-Met and EGFR with crizotinib and gefitinib, respectively, showed a similar trend as described above, with IC50 values for crizotinib ranging from 0.5 ± 0.02 μM to 6.5 ± 1.0 μM and generally being lower than that for gefitinib (to 0.22 ± 0.01 μM to 73.7 ± 7.7 μM) (Figure 2a). The IC50 values for the equimolar combination of crizotinib + gefitinib resembled that of crizotinib in cell lines wherein the difference between the IC50 values of gefitinib and crizotinib were 9-fold or higher. However, in MDA-MB-468, 4T1, PC3, DU145, UM22A, A549, H2170, VC8, VC8-MGMT, IGROV-1, SKOV3, and EFO-21 cells that exhibited IC50 values for crizotinib and gefitinib in the less than 6-fold difference range, the IC50 of the equimolar combination of crizotinib + gefitinib showed stronger potency than a single drug alone (Figure 2a). MGMT cells do not express the O6-methylguanine DNA methyltransferase (MGMT), a DNA repair enzyme that removes the cytotoxic O6-methyl group of guanine by transferring it to its internal cysteine residues. This is well illustrated by the average graphs shown in Figure 2b–g, where the response profile of gefitinib (Figure 2b) was marked by significantly higher IC50 values compared with those of crizotinib (Figure 2c). Moreover, the response profile for gefitinib + crizotinib (Figure 2d) resembled that of crizotinib alone. Where differences in IC50 values for gefitinib and crizotinib were less than 5-fold, the response profile of the combination (Figure 2g) appeared to produce enhanced potency when compared with each drug alone (Figure 2e–g).

#### 2.1.3. c-Met-c-Src Targeting

The analysis was extended to c-Met and c-Src targeting using crizotinib and dasatinib (a clinical c-Src/c-Abl inhibitor), respectively. The results showed that the IC50 of dasatinib ranged from 0.01 ± 0.0003 μM to 15.1 ± 1.8 μM and that of crizotinib ranged from 0.5 ± 0.02 μM to 6.5 ± 1.0 μM (Figure 3a). As previously observed, the IC50 of the equimolar combination of crizotinib + dasatinib resembled that of dasatinib in cell lines that demonstrated a 9-fold or higher difference between the IC50 of crizotinib and dasatinib. By contrast, in the MDA-MB-468 cell line wherein the difference between the IC50 values of crizotinib and dasatinib was 6-fold, the equimolar combination of the two drugs demonstrated 3- and 21-fold superior potency compared with crizotinib and gefitinib, respectively (Figure 3a).

#### 2.1.4. EGFR- or c-Met-DNA Targeting

The analysis was also carried out using a more divergent type of targeting involving one kinase inhibitor (e.g., EGFR or c-Met) and a DNA damaging agent. We determined the growth inhibitory potency of gefitinib (EGFR TKI) or crizotinib (c-Met inhibitor) alone or in equimolar combination with temozolomide (DNA alkylating agent) on a panel of cell lines (Figure 3c). The results showed that most of the cell lines were resistant to temozolomide, with IC50 values ranging from 430 ± 49 μM to > 800 μM. By contrast, all the cell lines demonstrated sensitivity to gefitinib or crizotinib, with IC50 values ranging from 0.3 ± 0.1 μM to 4.2 ± 1.1 μM for crizotinib and 4.7 ± 0.3 μM to 27.1 ± 1.6 μM for gefitinib. All cell lines exhibited more than a 30-fold difference in IC50 values between temozolomide and gefitinib or crizotinib. As for the kinase–kinase imbalanced targeting combinations, the trend is that the IC50 values for equimolar combinations of gefitinib + temozolomide or crizotinib + temozolomide resembled that of gefitinib alone or crizotinib alone, respectively (Figure 3b,c).

### 2.2. Parameterization of the Response Profiles

Overall, from our results, if we label IC50 values as γ, it appears that in vitro, when the difference in the IC50 of the less potent drug (γ1) and the more potent drug (γ2) is higher than 6-fold, the IC50 value for the equimolar combination referred to as γ3 is in the same range as γ2 (the more potent drug). However, if the difference in IC50 between γ1 and γ2 is less than or equal to 6-fold, which we consider to be a similar IC50 range, the equimolar combination exhibits superior potency when compared with each drug alone. We categorized the data into two groups based on the fold difference calculated as γ1/γ2; this term is referred to as κ. For imbalanced targeting (κ > 6), the average distribution of all the IC50 values of kinase inhibitor-1 (γ1) and kinase inhibitor-2 (γ2) across various cell lines showed a significant difference (*p* < 0.001). Consequently, the average IC50 for the equimolar drug combination (γ3) was in the same range as that of γ2 (IC50 of kinase inhibitor-2). In contrast to balanced targeting, there was no significant difference between γ1 and γ2 (*p* > 0.05) for data with κ ≤ 6 (Figure 4b).

### 2.3. Unimolecular Combinations

#### 2.3.1. EGFR-c-Src Targeting Combi-Molecules

In recent years, as part of our kinase–kinase targeting program, we have embarked upon the design and synthesis of EGFR-c-Src-targeting combi-molecules, which led to the development of a series of compounds, including AL660, AL690, AL692, AL739, and the optimized combi-molecule AL776 (see Figure 5a, structures shown in Appendix A) designed to induce tandem blockade of the two kinases, both as an intact structure and upon undergoing hydrolysis [19]. AL660, AL690, AL692, and AL739 behave as imbalanced kinase inhibitors due to their inherent ability to inhibit c-Src more potently than EGFR [19]. Here, we showed that all four combi-molecules were strong inhibitors of c-Src that blocked its phosphorylation at a concentration as low as 1 μM in whole cells (Figure 5b). In contrast, they were poor inhibitors of EGFR at the same concentration. They were unable to induce superior growth inhibitory potency compared with the equimolar combination of gefitinib + dasatinib in the NIH3T3 panel of cell lines (except for AL692 on NIH3T3-wt cells, which showed remarkable potency, possibly due to unspecific kinase binding) (Figure 5c,f). Optimization studies led to the identification of AL776, which not only exhibited potent EGFR and c-Src inhibitory potency in an in vitro kinase assay, but also in a whole cell immunoblot assay [19]. Consequently, we have a set of non-optimized and optimized EGFR-c-Src-targeting compounds that can be used to define criteria for predicting potency in comparison with equimolar combinations of gefitinib and dasatinib.

#### 2.3.2. Design, Synthesis, and Biological Potency of LP121, an EGFR-c-Met-Targeting Combi-Molecule

In order to explore situations wherein EGFR and c-Met are co-targeted with a combi-molecule, we developed LP121 (see structure in Appendix A), containing a carbonate linker bridging an EGFR and c-Met inhibitors. The results showed that LP121 could block both EGFR and c-Met, with an IC50 of 1 μM using an in vitro kinase assay (Figure 5d). In the 4T1 cell system, a cell line expressing both receptors, LP121 induced a dose-dependent inhibition of EGFR and c-Met phosphorylation following 2 h of treatment, indicating that the combi-molecule possessed dual-targeting properties (Figure 5e). Thus, we have in hand an EGFR-c-Met-targeting molecule that can be used to develop potency criteria in comparison with equimolar combinations.

### 2.4. Parameterization of Potency of Combi-Molecules: A New Parameter Ω as a Potency Index

The potency index of combi-molecules can only be defined in the context of a comparison with balanced equimolar combinations. Thus, as defined by the equation below, if γ4 is the IC50 of the combi-molecule, its potency index Ω can be calculated as a ratio of γ4 over the IC50 for an equimolar combination, γ3.
Ω = γ4/γ3

Thus, a combi-molecule can be considered effective only when Ω ≤ 1 in cells exerting balanced targeting through its two arms.

Analysis of the imbalanced combi-molecules AL660–AL739 (except for AL692 in wild type cells) gave Ω values far greater than 1 in cells expressing their target oncogenes, indicating that EGFR and c-Src are not adequately targeted by these combi-molecules (Figure 5b,c). Optimization of their structures gave AL776, which when evaluated in the NIH3T3 panel of cells, demonstrated superior potency compared with a single drug alone, but was unable to induce superior potency when compared with the equimolar combination of gefitinib + dasatinib (Ω > 1) (Figure 5c). However, given the dual inhibitory properties exerted by AL776, we further evaluated it in a panel of cancer cell lines, which like the NIH3T3 cells have previously demonstrated superior potency with the equimolar combination of gefitinib + dasatinib. The results showed that in both 22RV1 (prostate cancer) and A2780 (ovarian cancer) cells, AL776 was superior to or equal to the combination of gefitinib + dasatinib, with Ω = 0.4 in 22RV1 cells (IC50 = 3.7 ± 0.5 μM) and Ω = 1.0 in A2780 cells (IC50 = 2.2 ± 0.4 μM), indicating that this combi-molecule is dual targetingin these cells.

Given the potency of LP121 in blocking both c-Met and EGFR tyrosine kinase activity, we evaluated its growth inhibitory properties in cell lines exhibiting less than a 6-fold difference in IC50 values between gefitinib and crizotinib, as previously determined. The cell lines chosen included DU145, PC3 (prostate cancer), 4T1 (mouse mammary tumor cells), IGROV-1, SKOV-3, and EFO-21 (ovarian cancer), which exhibited sensitivity to the equimolar combination of gefitinib + crizotinib when compared with an individual drug alone. The results showed that among all the cell lines analyzed, LP121 showed superior activity compared with the equimolar combination in SKOV-3 cells (IC50 = 0.29 ± 0.04 μM and Ω = 0.7) and similar potency to the equimolar combination in EFO-21 cells (IC50 = 0.01 ± 0.001 μM and Ω = 1.0), indicating that LP121 is an effective combi-molecule against these cells.

## 3. Discussion

The complexity of signaling networks driving advanced cancers has given rise to several multi-targeted strategies, including the use of complex drug cocktails, hybrid or chimeric molecules (e.g., HDAC-tyrosine kinase inhibitors such as HDAC-EGFR/Her2, HDAC-PDGFR inhibitors, and microtubule disruptors) which are in preclinical/clinical stages of evaluation or currently being used in the clinic [37,38,39,40,41,42]. Within the context of developing single drugs with multi-targeted properties, our laboratory has specialized in synthesizing and developing a novel class of compounds termed “combi-molecules” that are designed to inhibit two distinct biological targets, both as an intact structure and/or to generate their potent inhibitory arms directed at the two targets upon hydrolysis. These biological targets (e.g., receptor tyrosine kinases EGFR, c-Met, and non-receptor tyrosine kinase c-Src) are known to synergistically potentiate each other’s effects in driving tumor growth and provide a strong rationale for developing a multi-targeted approach against their adverse effects. In theory, combi-molecules are designed to generate the molar equivalent of their two inhibitory arms upon undergoing hydrolysis in cells, thereby mimicking the effects of an equimolar combination of drugs [9,18,43,44,45]. The molar equivalent of activity is also assumed for hybrid molecules. Indeed, Loedige et al. [30] analyzed the potency of hybrid anti-malarial drugs in comparison with an equimolar combination of drugs representing the two moieties. While methods are available to assess the potency of an equieffective drug combination (i.e., the median-effect principle proposed by Chou-Talalay, which determines synergy, antagonism, and additive nature of drug combinations based on their combination index values), the same principles are not applicable to the outcome of equimolar drug combinations or unimolecular entities such as hybrid drugs or combi-molecules [32]. We thus sought to assess the overall potency of an equimolar combination of two drugs and a comparison with its unimolecular analog, using current and past data.

The general trend observed was that when the IC50 value of one drug was 6-fold higher than that of the other (fold difference of κ > 6), the IC50 of the equimolar combination of the two drugs resembled that of the more potent inhibitor. However, when the difference in IC50 values of the two drugs was equal to or less than 6-fold, the equimolar combination of the two drugs showed superior potency compared with a single drug alone, thereby exhibiting what we designate as “balanced targeting”. Our choice to define the κ threshold at 6-fold is based on the small sample size we have evaluated (10 drugs/combinations across 27 cell lines) and could be determined differently by other teams depending on their cell models and the nature of the drugs or combinations being tested.

The biological significance of these observations can be analyzed considering the dominant pathways that control cell growth and the ability to undergo apoptosis in human tumor cells. Being at the crossroad of multiple signaling pathways, blockade of c-Src with the potent inhibitor dasatinib led to IC50 values in the nanomolar range, which renders the blockade of other receptors (such as EGFR or c-Met) a minor contributor to the overall IC50 in combination. It is also important to also note that dasatinib exerts its action through inhibiting multiple other targets (polypharmacology), which could also be contributing towards its overall superior potency in certain cell lines [46]. As another example, HCC827 is a lung cancer cell line driven by activating EGFR mutations (e.g., del E746-A750 and L859R), and thus demonstrated potent inhibition (nM range IC50) by gefitinib. Not surprisingly, the IC50 of gefitinib in combination with dasatinib resembled that of gefitinib (Figure 1a). Clearly, in the case of an imbalanced response, equimolecular combinations are ineffective and seem to indicate that the cells are more dependent on a particular pathway.

In cells where balanced targeting is observed, perhaps the two targets synergize to promote growth and anti-apoptotic effects. Therefore, their tandem modulation leads to superior growth inhibitory potency compared with either drug alone. Importantly, these cell lines responding better to equimolar drug combinations are likely to benefit from hybrid/combi-molecules, as demonstrated in this study. Finally, if a combi-molecule achieves Ω < 1 (the potency index for unimolecular drugs calculated according to the equation Ω = γ4/γ3) in these cells, it can be considered worthy of further in vitro analysis and in vivo studies in xenograft models carrying the latter cell lines.

Steps toward defining potency criteria for combi-molecules are schematized in Figure 6. In summary, in vitro, our studies allow us to conclude the following:If two drugs are combined at an equimolar ratio, and one drug shows a 6-fold greater IC50 than the other (i.e., κ > 6), the overall effect will resemble the IC50 of the combination.If two drugs are combined at an equimolar ratio, and the IC50 of one drug is 6-fold or less than that of the other (i.e., κ ≤ 6), then the overall effect is superior to that of each individual drug, leading to balanced targeting.Under conditions of balanced targeting, a unimolecular combination (e.g., combi-molecule) is said to be effective if the IC50 of the combi-molecule is equal to or a fraction of the IC50 of the equimolar 2-drug combination (i.e., Ω ≤ 1).

## 4. Materials and Methods

### 4.1. Combi-Molecule Synthesis

The EGFR-c-Src combi-molecules AL660, AL690, AL692, AL739, and the EGFR-c-Met-targeting combi-molecule LP121 were synthesized according to the methods described in the Appendix A. AL776 was synthesized according to methods previously described in Rao et al., 2015 [19].

### 4.2. Cell Culture

The human cancer cell lines used in the present study include breast (MDA-MB-231, BT549, and MDA-MB-468), prostate (DU145, PC3, and 22RV1), lung (A549, A427, A427-MGMT, H2170, H1975, and HCC827), Chinese hamster lung cancer cells (VC8 and VC8-MGMT), ovarian (IGROV-1, SKOV-3, EFO-21, A2780, and OVCAR-3), head and neck (UM22A), NIH3T3 wild type (Wt), EGFR (Her14) and Her2 (Neu) transfected cells, and mouse mammary tumor 4T1 cells. MGMT cells do not express the O6-methylguanine DNA methyltransferase (MGMT), which is a DNA repair enzyme that removes the cytotoxic O6-methyl group of guanine by transferring it to its internal cysteine residues. The prostate cancer cells were a generous gift from Dr. Amina Zoubeidi (Vancouver Prostate Centre, Department of Urologic Sciences, University of British Columbia, Vancouver, BC, Canada). The 4T1, U87, and U87-MGMT cells were a generous gift from Dr. Thierry Muanza (Department of Oncology, Division of Radiation Oncology, Jewish General Hospital, Montreal, QC, Canada). The 4T1 cells were originally isolated by Dr. Fred Miller (Karmanos Cancer Institute, Detroit, MI, USA) [47]. The NIH3T3 panel of cells was a generous gift from Dr. Moulay Alaoui-Jamali (Lady Davis Institute for Medical Research Sir Mortimer B. Davis, Jewish General Hospital, Montreal, QC, Canada). The IGROV-1 cells were a generous gift from the Gustave Roussy Institute (Villejuif, France). The A2780 cells were purchased from Sigma Aldrich (Saint-Quentin-Fallavier, France). The V79 and V79-MGMT cells were kindly given by Dr. Bernd Kaina (Institute of Toxicology, University Medical Center, Mainz, Germany). The A427-MGMT cells were obtained from American Type Culture Collection (ATCC, Manassas, VA, USA) and transfected with MGMT in our lab [7]. The remaining cell lines were purchased from the ATCC. The H1975, HCC827, IGROV-1, SKOV-3, EFO-21, A2780, and OVCAR-3 cells were maintained in RPMI-1640 medium. The remaining cell lines were maintained in Dulbecco Modified Eagle’s Medium (DMEM). Both media were supplemented with 10% FBS, 10 mM HEPES, 2 mM L-glutamine, gentamycin sulfate, and fungizone (all reagents were purchased from Wisent Inc., St-Bruno, QC, Canada). The cells were grown in a humidified incubator with 5% CO_2_ at 37 °C.

### 4.3. Drug Treatment

Crizotinib was purchased from PharmaBlock USA, Inc. (Sunnyvale, CA, USA) and gefitinib from the Royal Victoria Hospital (Montreal, Canada) pharmacy and extracted from pills in our laboratory. Dasatinib was purchased from Ark Pharm Inc., Arlington Heights, IL, USA. Temozolomide was extracted form Temodal pills purchased form Merck/Schering Plough (Kenilworth, NJ, USA). All drugs were dissolved in DMSO to obtain a concentration of 40 mM or lower. Drug dilutions were carried out under sterile conditions using RPMI or DMEM (10% FBS) medium and the final concentration of DMSO never exceeded 1% (*v*/*v*).

### 4.4. Growth Inhibition Assay

Sulforhodamine B assay was used to measure growth inhibition in cells [48]. Cells were plated (5000–10,000 cells/well) and 24 h later treated with a dose range of single or combinations of drugs. After 5 days of treatment, cells were fixed in 50% trichloroacetic acid (TCA) for 2–3 h at 4 °C, washed 4 times under cold tap water, and stained with SRB (0.4%) for 2 h–overnight at room temperature. Plates were rinsed with 1% acetic acid and allowed to dry overnight, stained cells were dissolved in 10 mm Tris-Base, and the plates were read using a microplate reader ELx808 (492 nm). GraphPad Prism 6.0 (GraphPadSoftware, Inc., San Diego, CA, USA) was used for data processing. Each experiment was repeated at least twice, in triplicate.

### 4.5. In Vitro Kinase Assay

EGFR and c-Met in vitro kinase assays were carried out in 96-well plates (Nunc Maxisorp) coated with PGT (poly L-glutamic acid L-tyrosine, 4:1; Sigma Aldrich, St. Louis, MO, USA) and incubated at 37 °C for 48 h. PGT was the substrate to be phosphorylated by EGFR (Enzo Life Sciences Inc, Farmingdale, NY, USA; Signal Chem, Richmond, BC, Canada) or c-Met (BPS Bioscience, San Diego, CA, USA) in the presence of ATP (50 μm). Drugs (LP121, gefitinib, and crizotinib) were added, followed by 13.3 ng/well of isolated EGFR (0.1 μg/μL) or 32 ng/well of c-Met (0.75 μg/μL). The HRP-conjugated anti-phosphotyrosine antibody (Santa Cruz Biotechnology, Dallas, CA, USA) was used for phosphorylated substrate detection. The signal was developed using 3,3′,5,5′-tetramethylbenzidine peroxidase substrate (Kierkegaard and Perry Laboratories, Gaithersburg, MD, USA) and assessed using a microplate reader ELx808 at 450 nm (BioTek Instruments, Winusky, VT, USA). GraphPad Prism 6.0 (GraphPadSoftware, Inc., San Diego, CA, USA) was used for IC50 determination and each experiment was repeated at least twice, in duplicate.

### 4.6. Western Blot Analysis

NIH3T3-Her14 and 4T1 cells were plated and 24 h later rinsed twice with PBS and starved overnight using serum-free media. Cells were next treated with various concentrations of inhibitors, for 2 h, washed with PBS twice, and stimulated with 50 ng/mL EGF (NIH3T3-Her14) or EGF + HGF (4T1), each for 30 min at 37 °C. Western blot analysis was carried out according to methods previously described by Rao et al. [19]. Phosphotyrosine antibodies against EGFR (Y1068), c-Src (Y416), and c-Met (Y1234/1235), and also total c-Met and c-Src antibodies were purchased from Cell Signaling Technology, Danvers, MA, USA. Total EGFR and actin antibodies were purchased from Santa Cruz Biotechnology, Inc., Dallas, TX, USA. Immunoblot bands were visualized using Pierce™ ECL Western Blotting Substrate (Life Technologies Inc., Burlington, ON, Canada).

## Figures and Tables

**Figure 1 ijms-22-09569-f001:**
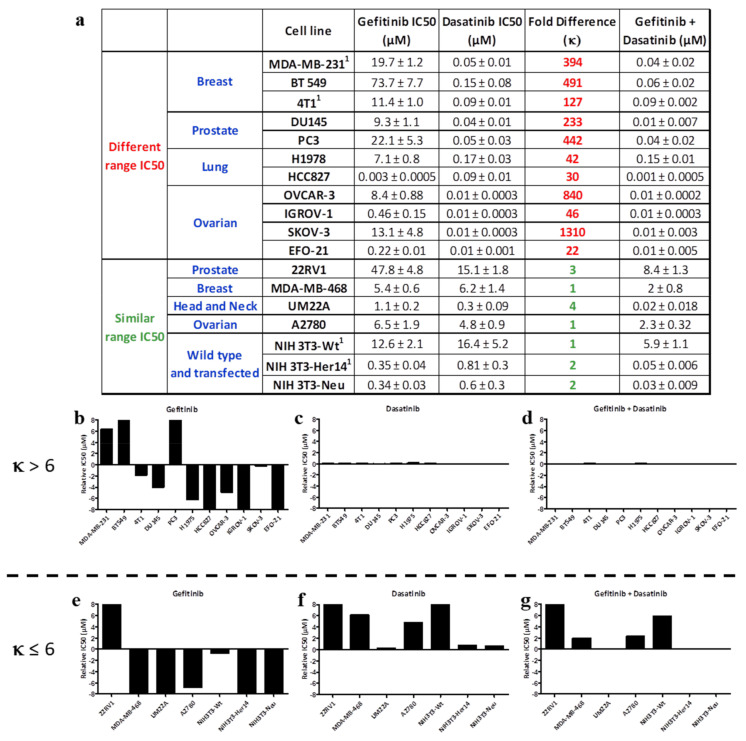
(**a**) IC50 values of tumor cell growth inhibition on a panel of human cancer cell lines using gefitinib (EGFR kinase inhibitor), dasatinib (c-Src kinase inhibitor), and their equimolar combination. The fold difference between the IC50 values of the two individual drugs is denoted by κ. (**b**–**g**) Relative IC50 values (IC50 of individual drug minus the average IC50 of the drug on the entire panel of cell lines) of gefitinib, dasatinib, and their equimolar combination for (**b**–**d**) κ > 6 and (**e**–**g**) κ ≤ 6. ^1^ Rao S. et al., 2015 [19].

**Figure 2 ijms-22-09569-f002:**
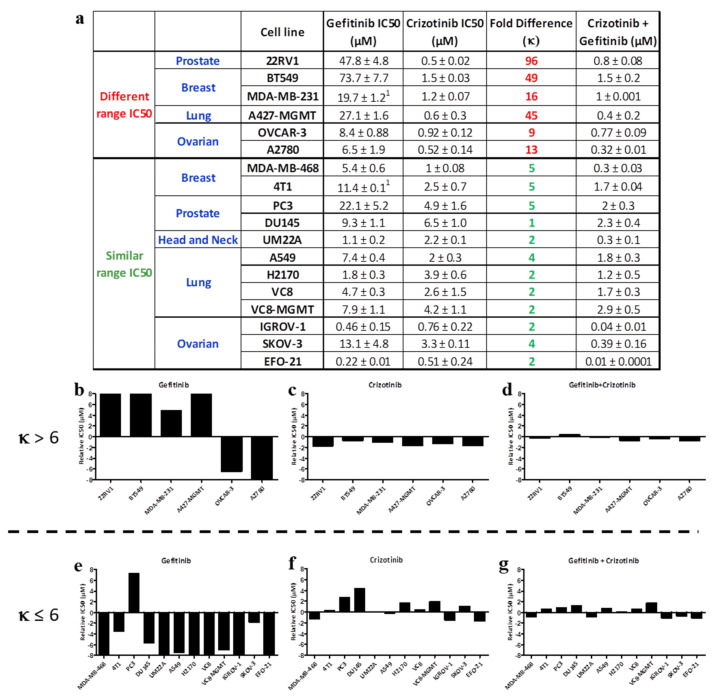
(**a**) IC50 values of tumor cell growth inhibition on a panel of human cancer cell lines using gefitinib, crizotinib (c-Met kinase inhibitor), and their equimolar combination. The fold difference between the IC50 values of the two individual drugs is denoted by κ. (**b**–**g**) Relative IC50 values (IC50 of individual drug minus the average IC50 of the drug on the entire panel of cell lines) of gefitinib, crizotinib, and their equimolar combination for (**b**–**d**) κ > 6, and (**e**–**g**) κ ≤ 6. ^1^ Rao et al. 2015 [19].

**Figure 3 ijms-22-09569-f003:**
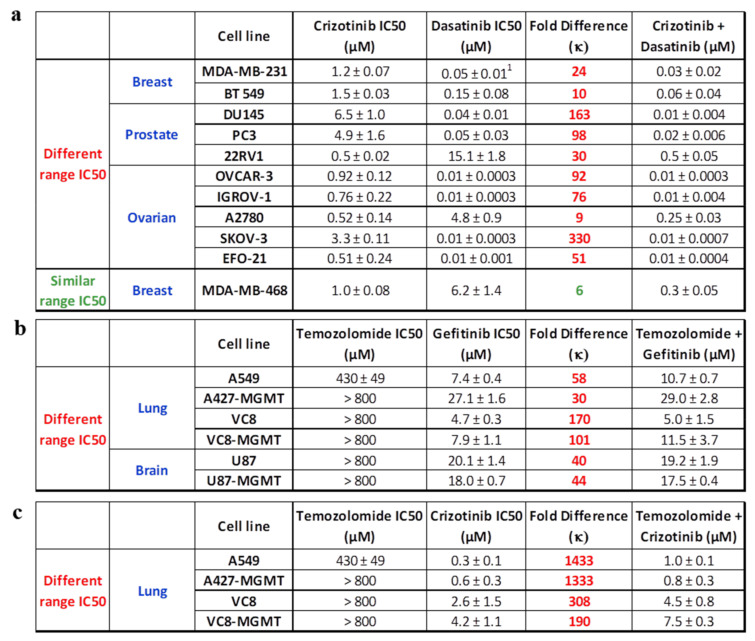
(**a**) IC50 values of growth inhibition assay targeting c-Met and c-Src using crizotinib, dasatinib, and their equimolar combination. (**b**) IC50 values of growth inhibition targeting EGFR and DNA using gefitinib, temozolomide (DNA alkylating agent), and their equimolar combination. (**c**) IC50 values of growth inhibition assay targeting c-Met and DNA using crizotinib, temozolomide, and their equimolar combination. The fold difference between the IC50 values of the two individual drugs is denoted by κ. ^1^ Rao et al. 2015 [19].

**Figure 4 ijms-22-09569-f004:**
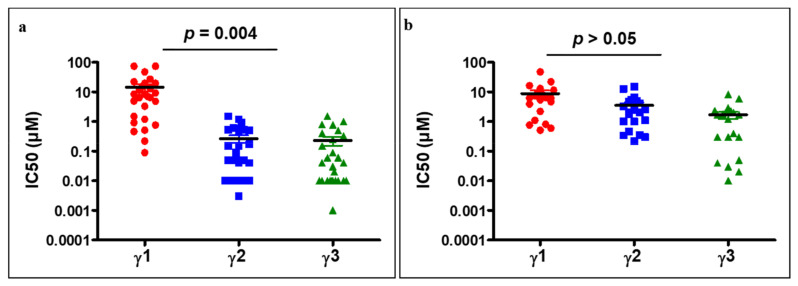
Comparing the distribution of IC50 values of kinase inhibitor-1 (γ1), kinase inhibitor-2 (γ2), and their equimolar combination (γ3) across all cell lines exhibiting a fold difference in IC50 of (**a**) > 6 and (**b**) ≤ 6. Note that γ1 > γ2 and κ represents the fold difference, which is calculated as γ1/γ2. Statistical analysis was carried out using unpaired two-tailed Student *t*-test.

**Figure 5 ijms-22-09569-f005:**
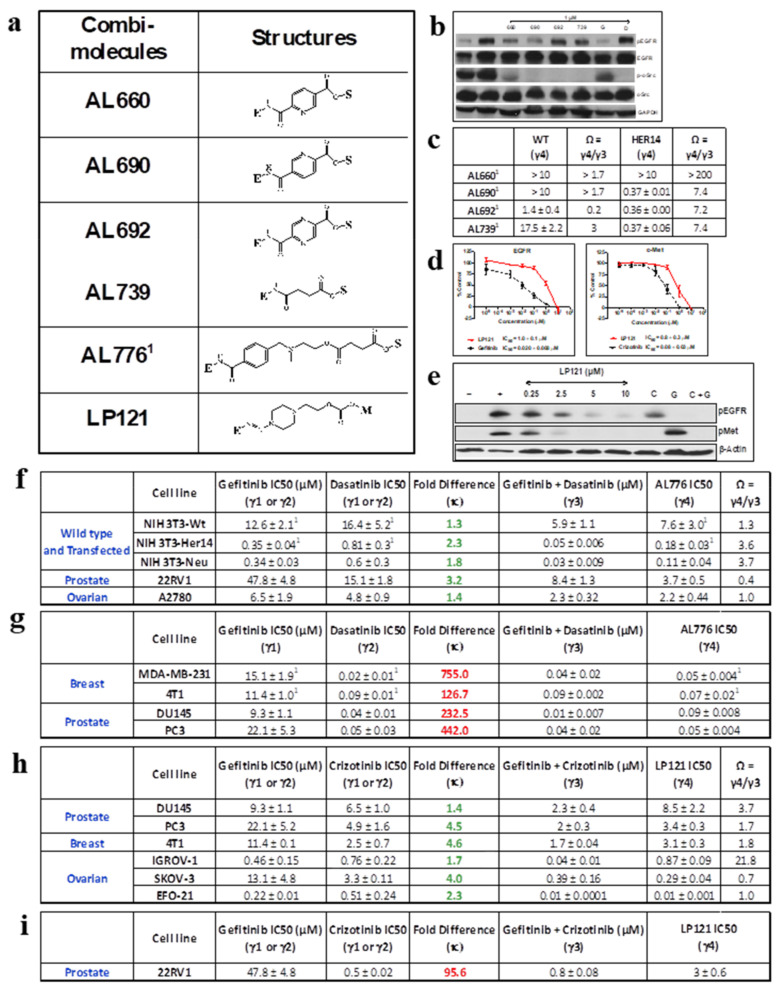
(**a**) Structures of EGFR-c-Src (AL660-AL776) and EGFR-c-Met (LP121) combi-molecules. E = EGFR targeting head, S = c-Src targeting head, and M = c-Met targeting head (structures shown in Appendix A and Appendix A). (**b**) Immunoblot analysis of inhibition of EGFR and c-Src phosphorylation by 1 μM dose of EGFR-c-Src-targeting combi-molecules (i.e., AL660, AL690, AL692, AL739) and the clinical inhibitors gefitinib (G) and dasatinib (D) in NIH3T3-Her14 (EGFR transfected) cells stimulated with 50 ng/mL of EGF. (**c**) IC50 values of growth inhibition and the combi-targeting effect, Ω (γ4/γ3) for imbalanced targeting combi-molecule on the NIH3T3-wild type and Her14 (EGFR transfected) cells. (**d**) EGFR and c-Met kinase inhibitory potency of LP121 using an in vitro kinase assay. (**e**) Target modulation by LP121, C—crizotinib (5 μM), G—gefitinib (5 μM), and C + G—an equimolar combination of crizotinib + gefitinib (5 μM each) in 4T1 cells using western blot analysis under conditions of EGF + HGF (50 ng/mL each) stimulation. (**f**,**g**) IC50 values of growth inhibition of the balance-targeted combi-molecule AL776 and the equivalent equimolar drug combination. Ω (γ4/γ3) was calculated for cell lines with κ ≤ 6. (**h**,**i**) IC50 values of growth inhibition of the balanced-targeted combi-molecule LP121 and the equivalent equimolar drug combination. Ω (γ4/γ3) was calculated for cell lines with κ ≤ 6. ^1^ Rao et al. 2015 [19].

**Figure 6 ijms-22-09569-f006:**
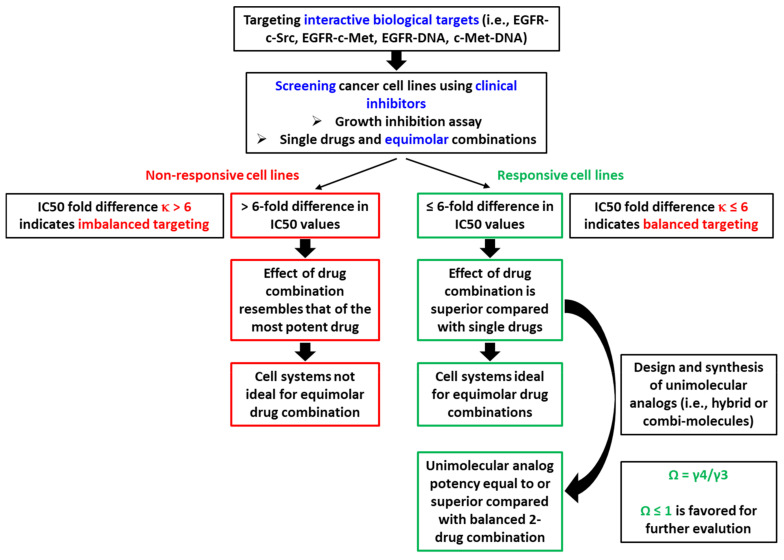
Flow chart summarizing the sequence of events leading to the design and synthesis of unimolecular analogs (i.e., hybrid drugs, chimeric molecules, and combi-molecules) as predicted on the basis of the fold difference between individual drugs (κ). γ1 = IC50 of drug 1, γ2 = IC50 of drug 2, γ3 = IC50 of equimolar combination of drug 1 + drug 2, γ4 = IC50 of unimolecular analog (e.g., combi-molecule), κ = γ1/γ2 where γ1 > γ2, and Ω = combi-targeting effect of unimolecular analog calculated using the equation Ω = γ4/γ3.

## Data Availability

Not applicable.

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
