# Peer review of "Quantitative Analysis of the Potency of Equimolar Two-Drug Combinations and Combi-Molecules Involving Kinase Inhibitors In Vitro: The Concept of Balanced Targeting"

_ijms, 2021, doi:10.3390/ijms22179569_

Round 1

Reviewer 1 Report

The paper by Rao et al.,  entitled "Quantitative Analysis of the Potency of Equimolar Two-Drug Combinations and Combi-Molecules Involving Kinase Inhibitors In Vitro: the Concept of Balanced Targeting" deals with the problem of assaying the combined effects of bioactive molecules acting simultaneously in equimolar doses in a protocols called " combi-molecules".  

The authors report simple mathematical models to interpret data newly obtained or available from the literature. The paper is carefully written and well supported by convincing results and interpretations. 

I recommend the publication without further modifications.

Author Response

We thank the reviewers for theirs comments.

Some changes have been made in the manuscript after comments from other reviewers and have improved the quality of the manuscript.

Reviewer 2 Report

Rao et al., presented potency indices for equimolar and dual targeted molecules. In the previous report, the combination index (CI) is used to define synergy. They proposed quantitative model based on simple mathematical equations for determining the degree of effectiveness of an equimolar drug combination. I find this manuscript is acceptable for publication in this journal. I suggest the authors make some corrections on the issues which is non-critical as follow: 

1.0μM -> 1.0 μM

40mM -> 40 mM

Author Response

We thank the reviewers for his comments.

We modified the manuscript according to his comment and added spaces when required (e.g. 1.0μM -> 1.0 μM).

Some changes have been made on the manuscript and has improved its quality after comments from other reviewers.

Reviewer 3 Report

The manuscript titled “Quantitative Analysis of the Potency of Equimolar Two-Drug Combinations and Combi-Molecules Involving Kinase Inhibitors In Vitro: the Concept of Balanced Targeting” proposes two new indeces ε and Ω to measure relative “potency” of equimolar drug combination and combi-molecules respectively. Combi-molecules in general are an interesting research direction and need attention and further studies. The comparison of potency of combi-molecules with that of corresponding equimolar mixtures is also actual. But the emphasis of the manuscript is put on the proposed novel indeces declared to guide researchers for achieving “balanced targeting”. I have some concerns with the mathematical part of the manuscript.

The ε index is found to be related to fold difference κ= γ1/γ2 since ε=[(γ3/γ1) + (γ3/γ2)]κ. The proposed index ε does not possess required numerical properties and thus is not a good measure of “potency”. First of all, it is directly proportional to γ3 because ε =[(γ3/γ1) + (γ3/γ2)]κ=[(1/γ1) + (1/γ2)]γ3κ. High γ3 values represent low in vitro effect, but will produce high ε values. The authors also studied correlation between ε and κ and have found strong correlation with R2=0.95. It follows that in real cases γ3 has no effect on ε values and these values are defined mostly by γ1/γ2 ratio, while both γ1 and γ2 are IC50 of arbitrarily chosen drugs. Thus the proposed ε index is invalid. Another index Ω is a straightforward “invention” since it is ratio of IC50 of the combi-molecule to the IC50 of the corresponding equimolar mixture. In this way the Ω index is not novel or interesting to a reader.

There is a lot of other minor mathematics and statistics-related issues. This undermines the quality of the manuscript.

In summary, I would not recomment this manuscript for publication.

Author Response

We thank the reviewer for his comments concerning our manuscript. The reviewer raised major concerns about the mathematics of the paper, stating the following;

Combi-molecules in general are an interesting research direction and need attention and further studies. The comparison of potency of combi-molecules with that of corresponding equimolar mixtures is also actual. But the emphasis of the manuscript is put on proposed novel indices declared to guide researchers for achieving “balanced targeting”. I have some concerns with the mathematical part of the manuscript.

The ε index is found to be related to fold difference κ= γ1/γ2 since ε=[(γ3/γ1) + (γ3/γ2)]κ. The proposed index ε does not possess required numerical properties and thus is not a good measure of “potency”. First of all, it is directly proportional to γ3 because ε =[(γ3/γ1) + (γ3/γ2)]κ=[(1/γ1) + (1/γ2)]γ3κ. High γ3 values represent low in vitro effect, but will produce high ε values. The authors also studied correlation between ε and κ and have found strong correlation with R2=0.95. It follows that in real cases γ3 has no effect on ε values and these values are defined mostly by γ1/γ2 ratio, while both γ1 and γ2 are IC50 of arbitrarily chosen drugs. Thus the proposed ε index is invalid. Another index Ω is a straightforward “invention” since it is ratio of IC50 of the combi-molecule to the IC50 of the corresponding equimolar mixture. In this way the Ω index is not novel or interesting to a reader.

We agree with the reviewer on the strong dependence of the equation on kappa. Therefore, to avoid confusion, equation 2 defining epsilon, was abandoned and corresponding data generated from the latter equation was removed. Kappa is presented as the primary exclusion criteria to consider a balanced targeting event in the cells.

While we agree to remove the epsilon on the basis that kappa is sufficient to guide on balanced targeting, omega must be kept as presented in the paper. It represents the main decision-making parameter to define the efficacy of a dual targeting combi-molecule. The rationale of omega (IC50 of combi-molecule/equimolar combinations) originates from a large number of papers by our group (22 cited in this manuscript), and is based on the simple fact that a combi-molecule is useful only if it can recapitulate or be more potent than an equimolar combination of 2 individual drugs. The novelty in this paper is that omega will only be applied to cells in which kappa is inferior or egal to 6 as defined empirically or simply when balanced targeting can be achieved.

Round 2

Reviewer 3 Report

The main remarks are taken into account. In the last version, the manuscript meets the requirements of the journal.

Author Response

We thank the reviewer for his comments and we are glad that our modifications filled his expectations.